# MAKING COHERENCE OUT OF NOTHING AT ALL: MEASURING EVOLUTION OF GRADIENT ALIGNMENT

## ABSTRACT

We propose a new metric ($m$-coherence) to experimentally study the alignment of per-example gradients during training. Intuitively, given a sample of size $m$, $m$-coherence is the number of examples in the sample that benefit from a small step along the gradient of any one example on average. We show that compared to other commonly used metrics, $m$-coherence is more interpretable, cheaper to compute ($O(m)$ instead of $O(m^2)$) and mathematically cleaner. (We note that $m$-coherence is closely connected to gradient diversity, a quantity previously used in some theoretical bounds.) Using $m$-coherence, we study the evolution of alignment of per-example gradients in ResNet and EfficientNet models on ImageNet and several variants with label noise, particularly from the perspective of the recently proposed Coherent Gradients (CG) theory that provides a simple, unified explanation for memorization and generalization [Chatterjee, ICLR 20]. Although we have several interesting takeaways, our most surprising result concerns memorization. Naïvely, one might expect that when training with completely random labels, each example is fitted independently, and so $m$-coherence should be close to 1. However, this is not the case: $m$-coherence reaches moderately high values during training (though still much smaller than real labels), indicating that over-parameterized neural networks find common patterns even in scenarios where generalization is not possible. A detailed analysis of this phenomenon provides both a deeper confirmation of CG, but at the same point puts into sharp relief what is missing from the theory in order to provide a complete explanation of generalization in neural networks.

## 1 INTRODUCTION

Generalization in neural networks trained with stochastic gradient descent (SGD) is not well-understood. For example, the generalization gap, i.e., the difference between training and test error depends critically on the dataset and we do not understand how. This is most clearly seen when we fix all aspects of training (e.g., architecture, optimizer, learning rate schedule, etc.) and vary only the dataset. In a typical experiment designed to test this, training on a real data set (e.g., ImageNet) leads to a relatively small generalization gap, whereas training on randomized data (e.g., ImageNet with random labels) leads to a much larger gap (Zhang et al., 2017; Arpit et al., 2017).

The mystery is that in both cases (real labels and random) the training accuracy is close to 100% which implies that the network and the learning algorithm have sufficient effective capacity (Arpit et al., 2017) to memorize the training sets, i.e., to fit an arbitrary mapping from the input images to labels. But, what then, is the mechanism that from among all the maps consistent with the training set, allows SGD to find one that generalizes well (when such a well-generalizing map exists)?

This question has motivated a lot of work (see e.g., Zhang et al. (2017); Arpit et al. (2017); Bartlett et al. (2017); Kawaguchi et al. (2017); Neyshabur et al. (2018); Arora et al. (2018); Belkin et al. (2019); Rahaman et al. (2019)) but no satisfactory answer has emerged. As Nagarajan & Kolter (2019) point out, traditional approaches based on uniform convergence may not suffice, and new ideas are needed. A promising line of attack is via algorithmic stability Bousquet & Elisseeff (2002), but traditional stability analysis of SGD (e.g., Hardt et al. (2016); Kuzborskij & Lampert (2018)) does not account for the dataset, and without that, one cannot hope to get more than a vacuous bound.

Recently, a new approach called Coherent Gradients (CG) has been proposed that takes into account the training dataset in reasoning about stability (Chatterjee, 2020; Zielinski et al., 2020). By analogy to Random Forests which also show dataset dependent generalization, CG posits that neural networks try to extract commonality from the dataset during the training process.

The key insight is that, since the overall gradient for a single step of SGD is the sum of the per-example gradients, it is strongest in directions that reduce the loss on multiple examples if such directions exist. Intuitively, at one extreme, if all the per-example gradients are aligned we get perfect stability (since dropping an example does not affect the overall gradient) and thus perfect generalization.

At the other extreme, if all the per-example gradients are pairwise orthogonal, we get no stability (since dropping an example eliminates any descent along its gradient), and thus pure memorization. This can be seen, for example, when trying to fit a linear model $y = w \cdot x$ to the following dataset under the usual mean squared error loss:

| $i$ | $x_i$ | $y_i$ |
|---|---|---|
| 0 | $\langle 1, \quad 0, \quad 0, \quad 0 \rangle$ | 1 |
| 1 | $\langle 0, -1, \quad 0, \quad 0 \rangle$ | $-1$ |
| 2 | $\langle 0, \quad 0, -1, \quad 0 \rangle$ | $-1$ |
| 3 | $\langle 0, \quad 0, \quad 0, \quad 1 \rangle$ | 1 |

Thus CG provides a simple, unified explanation for both memorization and generalization. However, at the same time, CG leads to some basic empirical questions:

1. *What does the alignment of per-example gradients, i.e.,* coherence *look like in practice?*

   As was noted in Chatterjee (2020), we expect a real dataset to have more coherence than a dataset with random labels, but how big is this difference quantitatively? Is coherence in the random label case like that in the pairwise orthogonal case described above? How does it vary with layer or architecture?

2. *Is the coherence constant throughout training, or does it vary? If so, how?*

   The key insight of CG (as described above) is a point-in-time observation, but in order to get a full picture of generalization we need to analyse the entire training trajectory. For example, one might imagine that as more and more training examples are fitted, coherence decreases, but is it possible for it to increase in the course of training?

In this paper, we propose a new metric called $m$-coherence to experimentally study gradient coherence. The metric admits a very natural intuitive interpretation that allows us to gain insight into the questions above. While we confirm our intuitions in many cases, we also find some surprises. These observations help us formulate more precisely what is missing from the CG explanation for generalization, and thus point the way to future work in this direction.

## 2 Prior Work on Metrics for Experimentally Measuring Coherence

**Pairwise Dot Product.** An obvious starting point to quantify the alignment or coherence of a set of gradients is their average pairwise dot product. Since this has a nice connection to the loss function, we start by reviewing the connection, and also set up notation in the process.

Formally, let $\mathcal{D}(z)$ denote the distribution[1] of examples from a finite[2] set $Z$, and assume without loss of generality that $\mathrm{support}(\mathcal{D}) = Z$. For a network with $d$ trainable parameters, let $\ell_z(w)$ be the loss for an example $z \sim \mathcal{D}$ for a parameter vector $w \in \mathbb{R}^d$. For the learning problem, we are interested in minimizing the expected loss $\ell(w) := \mathbb{E}_{z \sim \mathcal{D}}[\ell_z(w)]$. Let $g_z := [\nabla \ell_z](w)$ denote the gradient of the loss on example $z$, and $g := [\nabla \ell](w)$ denote the overall gradient. From linearity, we have,

$$g = \mathop{\mathbb{E}}_{z \sim \mathcal{D}} [\, g_z \,]$$

---

[1] We would like to quantify gradient coherence for both populations and samples. Therefore, $\mathcal{D}$ can either be a population distribution (typically unknown) or a sample (i.e., empirical) distribution.

[2] We assume finiteness for simplicity since it does not affect generality for practical applications.

Now, suppose we take a small descent step $h = -\eta g$ (where $\eta > 0$ is the learning rate). From the Taylor expansion of $\ell$ around $w$, we have,

$$\ell(w + h) - \ell(w) \approx g \cdot h = -\eta \, g \cdot g = -\eta \underset{z \sim \mathcal{D}}{\mathbb{E}} [ \, g_z \, ] \cdot \underset{z \sim \mathcal{D}}{\mathbb{E}} [ \, g_z \, ] = -\eta \underset{z \sim \mathcal{D}, z' \sim \mathcal{D}}{\mathbb{E}} [g_z \cdot g_{z'}] \quad (1)$$

where the last equality can be checked with a direct computation. Thus, the following are approximately equivalent:

- reduction in loss (due to a small step) divided by the learning rate,
- squared $\ell^2$ norm of the expected gradient, and,
- expected pairwise dot product (where the expectation is over *all* pairs).

**Example.** (Chatterjee, 2020) Consider a sample with $m$ examples $z_i$ where $1 \le i \le m$. Let $g_i$ be the gradient of $z_i$ and further that $\|g_i\| = \|u\|$ for some $u$. If all the $g_i$ are the same, then $g \cdot g = \|u\|^2$. However, if they are pairwise orthogonal, i.e., $g_i \cdot g_j = 0$ for $i \ne j$, then $g \cdot g = \frac{1}{m} \|u\|^2$. □

As this illustrates, the average expected dot product can vary significantly depending on the coherence. However, as a metric for coherence it is rather fragile. For example, just re-scaling the loss can drastically alter the value of the metric. Therefore, it can only be used to reason about coherence in very limited settings. For e.g., Chatterjee (2020); Zielinski et al. (2020) use it to verify that adding increasing amounts of label noise to a dataset reduces coherence but in order to do so they keep everything else the same, and limit their considerations to the start of training. But, to study the evolution of coherence, even over a single training run requires normalization since the magnitude of the gradients changes significantly in the course of training (e.g., see Appendix B).

**Stiffness.** Fort et al. in their preprint (2020) study two variants of the average pairwise dot product that they call *sign stiffness* and *cosine stiffness*. In our notation these are

$$S_{\text{sign}} := \underset{\substack{z \sim \mathcal{D}, z' \sim \mathcal{D} \\ z \ne z'}}{\mathbb{E}} [ \, \text{sign}(g_z \cdot g_{z'}) \, ] \text{ and } S_{\cos} := \underset{\substack{z \sim \mathcal{D}, z' \sim \mathcal{D} \\ z \ne z'}}{\mathbb{E}} \left[ \frac{g_z}{\|g_z\|} \cdot \frac{g_{z'}}{\|g_{z'}\|} \right].$$

These are meant to capture how a small gradient step based on one input example affects the loss on a *different* input example. Although Fort et al. do not describe why they choose to transform the gradients in these specific ways, we expect it is to normalize the dot product so that it can be tracked in the course of training. In their experience, they found sign stiffness to be more useful to analyze stiffness between classes whereas cosine stiffness was more useful within a class.

**Gradient Confusion.** Sankararaman et al. in their preprint (2019) introduce the notion of a gradient confusion bound. The *gradient confusion bound* is $\zeta \ge 0$ if for all $z, z' \in Z$ and $z \ne z'$, we have, $g_z \cdot g_{z'} \ge -\zeta$. They use this concept to study theoretically the convergence rate of gradient descent, but in their experimental results they measure the minimum cosine similarity between gradients, i.e.,

$$\underset{\substack{z \in Z, z' \in Z \\ z \ne z'}}{\min} \left[ \frac{g_z}{\|g_z\|} \cdot \frac{g_{z'}}{\|g_{z'}\|} \right]$$

We note that the non-linearities (and to a lesser extent the $z \ne z'$ restriction) make it hard to tie stiffness or minimum cosine similarity to what happens during training; specifically, to the change in the loss function as a result of a gradient step which is the expectation over *all* per-example gradients.

## 3 A New Metric for Coherence

The key insight behind our proposal is that there is a natural scaling factor that can be used to normalize the expected dot product of per-example gradients (i.e., the quantity in (1)) that preserves the connection to the loss. Consider the Taylor expansion of each individual loss $\ell_z$ around $w$ when we take a small step $h_z$ down *its* gradient $g_z$:

$$\ell_z(w + h_z) - \ell_z(w) \approx g_z \cdot h_z = -\eta \, g_z \cdot g_z$$

Taking expectations over $z$ we get,

$$\underset{z \sim \mathcal{D}}{\mathbb{E}} [\ell_z(w + h_z) - \ell_z(w)] = -\eta \underset{z \sim \mathcal{D}}{\mathbb{E}} [g_z \cdot g_z] \quad (2)$$

The quantity in (2) has a simple interpretation: It is the reduction in the *overall* loss $\ell$ if each example $\ell_z$ could be optimized independently. As might be expected intuitively, it is a lower bound on the quantity in (1) and is tight when all the per-example gradients are identical. We prove this formally in §4. Thus, it serves as a natural scaling factor for the expected dot product, and we obtain a normalized metric for coherence (denoted by $\alpha$) from (1) and (2):

$$\alpha := \frac{\ell(w+h) - \ell(w)}{\mathbb{E}_{z\sim\mathcal{D}}[\ell_z(w+h_z) - \ell_z(w)]} = \frac{\mathbb{E}_{z\sim\mathcal{D}, z'\sim\mathcal{D}}[g_z \cdot g_{z'}]}{\mathbb{E}_{z\sim\mathcal{D}}[g_z \cdot g_z]} = \frac{\mathbb{E}_{z\sim\mathcal{D}}[g_z] \cdot \mathbb{E}_{z\sim\mathcal{D}}[g_z]}{\mathbb{E}_{z\sim\mathcal{D}}[g_z \cdot g_z]} = \frac{\mathbb{E}_{z\sim\mathcal{D}}[g_z \cdot g]}{\mathbb{E}_{z\sim\mathcal{D}}[g_z \cdot g_z]} \quad (3)$$

*Thus, $\alpha$ is the change in the overall loss due to a small gradient step as a fraction of the maximum possible change in loss if each component of the loss could be optimized independently.*

As noted before, $0 \le \alpha \le 1$, and the maximum is achieved when all the gradients are identical, and the minimum is achieved when the expected gradient is 0, i.e., a stationary point is reached.

**A natural scale for $\alpha$.** Once again, consider a sample with $m$ examples $z_i$ where $1 \le i \le m$. Let $g_i$ be the gradient of $z_i$. Suppose further that the $g_i$ are pairwise orthogonal i.e. $g_i \cdot g_j = 0$ for $i \ne j$. It is easy to check that $\alpha = 1/m$. For a sample of size $m$, we call this value of $\alpha$ the *orthogonal limit*.

Since in the orthogonal case, each example is optimized independently, going down the expected gradient is $1/m$ times as slow as optimizing each independently. If the gradients are better aligned, we expect them to help each other resulting in an $\alpha$ greater than the orthogonal limit.

**Example (Commonality).** For $1 \le i \le m$, suppose each $g_i$ has a common component $c$ and an idiosyncratic component $u_i$, i.e., $g_i = c + u_i$ with $u_i \cdot u_j = 0$ for $1 \le j \le m$ and $j \ne i$; $u_i \cdot c = 0$; and say, $u_i \cdot u_i = \|u\|^2$ for some $u$. It is easy to see that $\alpha$ in this case is $\frac{1}{m}[1 + (m-1) \cdot f]$ where $f = \|c\|^2 / (\|c\|^2 + \|u\|^2)$. $\qquad\square$

These examples along with the observation that $0 \le \alpha \le 1$ suggests a more evocative (even if less accurate and less general) interpretation: In a given sample, $\alpha$ is the average fraction of examples that each example helps or supports. Thus, when analyzing experimental data, for a sample of size $m$, it is convenient to define a new quantity $m$-coherence as follows:

$$m\text{-coherence} := m \cdot \alpha = m \cdot \frac{\mathbb{E}_{z\sim\mathcal{D}, z'\sim\mathcal{D}}[g_z \cdot g_{z'}]}{\mathbb{E}_{z\sim\mathcal{D}}[g_z \cdot g_z]}$$

Thus $m$-coherence in the orthogonal limit is 1 and in the identical case is $m$. *Intuitively, $m$-coherence of a sample is the number of examples (including itself) that any one example helps on average.*

**Advantages.** $\alpha$ and $m$-coherence have several advantages over the metrics discussed in §2:

- **Computational Efficiency.** For a sample of size $m$, due to (3), $\alpha$ can be computed exactly in $O(m)$ time in contrast to $O(m^2)$ time required for stiffness and cosine dot products. Furthermore, it can be computed in a streaming fashion by keeping two running sums, so the per-example gradients need not be stored. Thus, in our experiments we are able to use sample sizes a couple of orders of magnitude higher than those in Fort et al. (2020) and Sankararaman et al. (2019).

- **Mathematical Simplicity.** We believe our definition is cleaner mathematically. This allows us to reason about the metric more easily. For example,

  1. We can show that the coherence of minibatch gradients is greater than that of individual examples (Corollary 3.1). Therefore, care must be taken if minibatch gradients are used in lieu of example gradients in computing coherence (e.g., as in Sankararaman et al. (2019)).
  2. Explicitly ruling out $z \ne z'$ as in done in stiffness and cosine similarity to eliminate self-correlation is unnatural and can get tricky in practice due to near-duplicates or multiple examples leading to same or very similar gradients. We obtain meaningful values without imposing those conditions, but if one insists on removing self-correlations, then subtracting $1/m$ from $\alpha$ or 1 from $m$-coherence is a more principled way to do it.
  3. The non-linearities in stiffness and cosine similarity amplify small per-example gradients potentially overstating their importance, and lead to a discontinuity (or undefined behavior) with zero gradients. However, we can cleanly account for the effect of negligible gradients in our observations (e.g., see Lemma 4).

- **Interpretability.** Finally, as discussed in detail above, they are normalized and yet easily interpretable due to the natural connection with loss.

**Prior Work on Gradient Diversity.** While writing this paper we discovered that the reciprocal of $\alpha$ appears in the theory literature as *gradient diversity*. This was used by Yin et al. (2018) in theoretical bounds to understand the effect of mini-batching on convergence of SGD. (A similar result appears for least squares regression in Jain et al. (2018).) They show that the greater is the gradient diversity, the more effective are large mini-batches in speeding up SGD. Although they support their theoretical analysis with experiments on CIFAR-10 (where they replicate $1/r$ of the dataset $r$ times and show that greater the value of $r$ less the effectiveness of mini-batching to speed up) they never actually measure the gradient diversity in their experiments (or further study its properties). Also, note that for our purposes $\alpha$ is a better choice than $1/\alpha$ – not just because coherence rather than incoherence is what leads to generalization – but also since the latter can diverge: $g$ can be 0 without all $g_z$ being zero (e.g., at the end of training in an under-parameterized setting).

## 4    A MORE GENERAL SETTING FOR COHERENCE AND SOME BASIC FACTS

Our notion of coherence is not specific to gradients (or optimization) but extends naturally to vectors in Euclidean spaces. Let $\mathcal{V}$ be a probability distribution on a collection of $m$ vectors in an Euclidean space. In accordance with (3), we define the *coherence* of $\mathcal{V}$ (denoted by $\alpha(\mathcal{V})$) to be

$$\alpha(\mathcal{V}) = \frac{\underset{v \sim \mathcal{V}, v' \sim \mathcal{V}}{\mathbb{E}} [v \cdot v']}{\underset{v \sim \mathcal{V}}{\mathbb{E}} [v \cdot v]} \tag{4}$$

Note that $\mathbb{E}[v \cdot v] = 0$ implies $\mathbb{E}[v \cdot v'] = 0$. In what follows, we ignore the technicality of the denominator being 0 by always assuming that there is at least one non-zero vector in the support of $\mathcal{V}$ (which also held in our experiments). We list some basic facts and the proofs are in Appendix A.

**Theorem 1** (Boundedness). *We have $0 \leq \alpha(\mathcal{V}) \leq 1$. In particular, $\alpha(\mathcal{V}) = 0$ iff $\mathbb{E}_{v \sim \mathcal{V}}[v] = 0$ and $\alpha(\mathcal{V}) = 1$ iff all the vectors are equal.*

**Lemma 2** (Scale Invariance). *For non-zero $k \in \mathbb{R}$, let $k\mathcal{V}$ denote the distribution of the random variable $kv$ where $v$ is drawn from $\mathcal{V}$. We have $\alpha(k\mathcal{V}) = \alpha(\mathcal{V})$.*

**Theorem 3** (Stylized mini-batching). *Let $v_1, v_2, .., v_k$ be $k$ i.i.d. variables drawn from $\mathcal{V}$. Let $\mathcal{W}$ denote the distribution of the random variable $w = \frac{1}{k} \sum_{i=1}^{k} v_i$. We have,*

$$\alpha(\mathcal{W}) = \frac{k \cdot \alpha(\mathcal{V})}{1 + (k-1) \cdot \alpha(\mathcal{V})} \tag{5}$$

**Corollary 3.1** (Minibatch amplification). *$\alpha(\mathcal{W}) \geq \alpha(\mathcal{V})$ with equality iff $\alpha(\mathcal{V}) = 0$ or $\alpha(\mathcal{V}) = 1$.*

**Remark.** This formulation provides a nice perspective on the type of results proved in Yin et al. (2018) and Jain et al. (2018). When $\alpha \ll 1/k$ but non-zero (i.e., we have high gradient diversity), creating mini-batches of size $k$ increases coherence almost $k$ times. But, when $\alpha \approx 1$ (i.e., low diversity) there is not much point in creating mini-batches since there is little room for improvement.

**Lemma 4** (Effect of zero gradients). *If $\mathcal{W}$ denotes the distribution where with probability $p > 0$ we pick a vector from $\mathcal{V}$ and with probability $1 - p$ we pick the zero vector then $\alpha(\mathcal{W}) = p \cdot \alpha(\mathcal{V})$.*

**Example (Coherence Reduction).** If we add $k$ zero gradients to the collection of gradients constructed in the example of §3 (Commonality), using Lemma 4, we get,

$$\alpha = \frac{m}{m+k} \cdot \frac{1}{m} \left[ 1 + (m-1) \cdot f \right] = \frac{1}{n} \left[ 1 + (n-k-1) \cdot f \right]$$

where $n = m + k$ is the size of this new sample. For a fixed $n$, as $k$ increases, $\alpha$ decreases going down to $1/n$ (the orthogonal limit) when all but one vector in the sample is zero, i.e., $k = n - 1$.

## 5    EXPERIMENTAL RESULTS

As stated in the introduction, our objective in developing $m$-coherence is to better understand the recently proposed CG explanation for generalization (Chatterjee, 2020; Zielinski et al., 2020). We

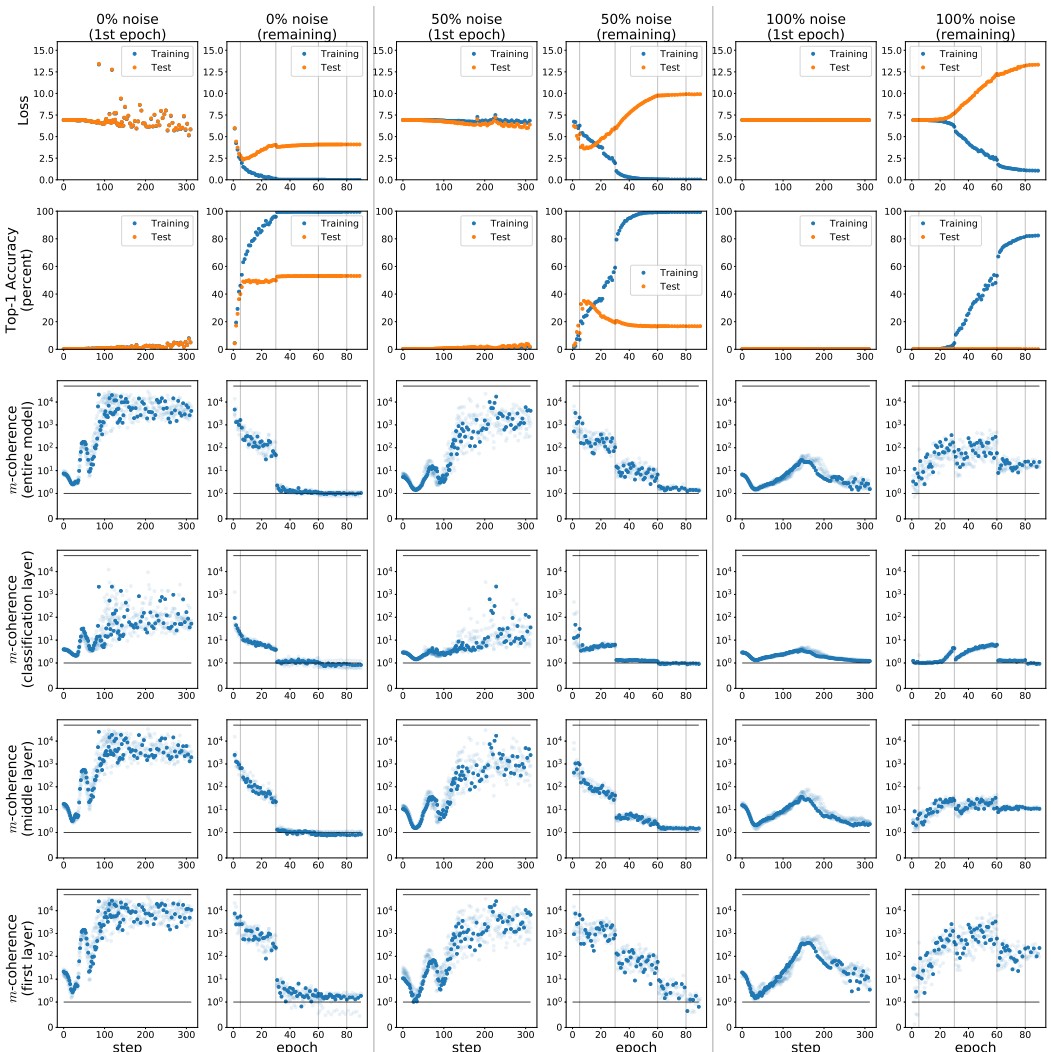

Figure 1: The evolution of alignment of per-example gradients of a ResNet-18 during training as measured with $m$-coherence on samples of size $m = 50{,}000$ on 3 variants of ImageNet with different amounts of label noise. Our main finding is that coherence not only decreases in the course of training (as expected from Lemma 4 when examples get fit), but it also increases. A high peak is reached rapidly with real labels (in the first 100 steps) and a much lower peak is reached slowly with random labels (over many epochs). Horizontal lines for $m$-coherence are shown at 1 (the orthogonal limit) and at $m$. Vertical lines indicate sharp reductions in learning rate. Light dots show the results of 4 other runs to understand sensitivity w.r.t. randomness in initialization and mini-batch construction.

now measure $m$-coherence in a setting very similar to that used by Zielinski et al. (2020) for studying memorization and generalization.

**Methodology.** We train ResNet-18 (He et al., 2016) models on ImageNet with original labels (0% noise), and two derived datasets: one with half the training labels randomized (50% noise), and another with all the training labels randomized (100% noise).[3] We use SGD with momentum (0.9), a batch size of 4096, and the learning rate schedule proposed in Goyal et al. (2017). We turn off augmentation and weight decay to observe memorization in the noisy cases within a reasonable number of steps. For each dataset, we track $m$-coherence on a random (but fixed) set of $m = 50{,}000$ *training* examples.

Figure 1 shows the data from our experiments. Each column corresponds to a different experiment and the rows show loss, accuracy, and $m$-coherence for the entire model and some specific layers.

---

[3]We use the original ImageNet validation set as our test set in all cases.

**Real Labels.** Our first experiment (shown in the *second* column of Figure 1 for reasons that will become clear shortly) measures the $m$-coherence (row 3) for training with 0% noise, i.e., the real ImageNet labels at the end of each epoch. The coherence at the end of the first few epochs is very high, between $10^3$ and $10^4$ and it decreases as more training examples get fit. We note that although there is some fluctuation in the coherence, it stays high (above $10^2$ and often above $10^3$) until well after the accuracy crosses the 50% mark. It settles at 1 after all the examples are fit.

The high coherence in early epochs agrees well with the intuition from CG that real datasets have good per-example gradient alignment since that is what is necessary for good generalization as per the theory. The subsequent decrease in coherence in the course of training is expected from Lemma 4 under the assumption that the gradients of fitted examples become small.

**Random Labels.** Our second experiment (column 6) shows that with random labels, the coherence in the first few epochs is low (between 1 and 10). It increases steadily until it reaches a peak in epochs 40 to 60 (above $10^2$ but less than $10^3$) followed by a decrease.

The low coherence (near the orthogonal limit) in the first few epochs agrees well with CG as discussed in the introduction but the subsequent increase is surprising (though not in contradiction with CG as discussed later). The increase is not small since at its peak each example is helping hundreds of other examples (though it is well below the $10^4$ peak seen with real labels). Once again, as examples get fitted, coherence decreases as expected from Lemma 4, though not back down to 1, likely since our training only goes on till about 80% accuracy is reached.

The increase in one case and not the other leads to a natural question with implications about the dynamics of SGD: *Is the evolution of coherence fundamentally different between the well-generalizing case (real labels) and the memorization case (random labels)?*

**Coherence in the First Epoch.** To study this question, we took a closer look at the first epoch. We recorded $m$-coherence at initialization (i.e., before the first step) and, thereafter, for every other step in the epoch. Since this requires computing the per-example gradients for 50K examples every 2 steps, this was our most computationally expensive experiment taking a day per run using TPUs. The results are shown in columns 1 (real labels) and 5 (random).

We find that in both cases coherence starts from about the same low point and starts rising. The rise is much faster for real labels than for random labels. We ran additional experiments with 50% noise (column 3) and with 25% and 75% noise to confirm that the slope with which coherence increases depends inversely on the amount of label noise (see Appendix B).

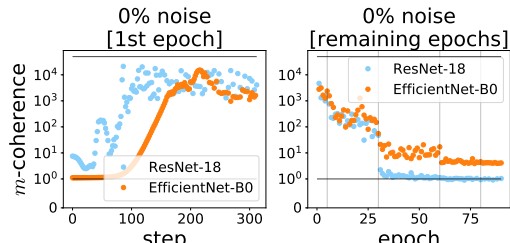

**The Overall Evolution.** If we combine the data from the steps of the first epoch with the rest of training (e.g., in row 3 we jointly view columns 1 and 2), we find a remarkably consistent pattern across all noise levels: $m$-coherence follows a broad parabolic trajectory (albeit with some local variation and noise) where it starts small, increases to a peak, and then goes back down. Thus, there is always an initial increase in coherence, just on different timescales.

Figure 2: The early trajectory of an EfficientNet-B0 model shows a similar increase as ResNet. The overall trajectories are also similar, and it is interesting to note that we get similar values for coherence although the two architectures are very different.

*From this point of view, the evolution of coherence in the memorization case does not look fundamentally different from that in the well-generalizing case.*

**Impact of Layers.** The bottom 3 rows of Figure 1 shows coherence by layer for 3 illustrative layers (the first convolution layer, a convolution layer in the middle, and the final fully connected layer). Although the specific values are different across the layers,[4] we notice that the broad trajectory

---

[4]Since, the interpretability of $m$-coherence allows for meaningful comparisons between layers, we can get some additional insight into the dynamics of training by studying these values. We do so in Appendix B.

observed for the coherence of the entire model holds for each individual layer. Thus, the trajectory (and in particular the increase) is not driven by one specific layer.

**Impact of Architecture.** We studied one other architecture, EfficientNet (Tan & Le, 2019), and found a similar increase in coherence and indeed a similar broad trajectory (Figure 2 and Appendix B).

**Coherence on Test Set.** For completeness, we also measured the coherence on $m$ examples not used for training (drawn from the ImageNet validation set). We defer the discussion to the Appendix.

**Reconciliation with Other Studies.** Although a direct comparison with the results in Fort et al. (2020) and Sankararaman et al. (2019) is not possible due to differences in metrics, sample sizes, datasets (ImageNet v/s CIFAR), and effects studied; in as much as they can be compared, we did not find contradictions. Please see Appendix B for more details.

# 6    DISCUSSION AND FUTURE WORK

In this paper, we have presented $m$-coherence, a mathematically clean and easily interpretable metric to study the evolution of the alignment of per-example gradients in the course of training. As stated in the introduction, our motivation in developing this metric is to better understand the recently proposed CG explanation for generalization (Chatterjee, 2020; Zielinski et al., 2020).

As noted in Sections 2 and 3, in contrast to unnormalized metrics such as the expected pairwise dot products of per-example gradients (which are difficult to compare across different training steps or architectures), or previously proposed normalized metrics such as stiffness or confusion (which are difficult to interpret due to ad hoc normalization that breaks the connection with the loss), the natural normalization employed in $m$-coherence allows us to directly interpret the value of $m$-coherence as the number of examples (including itself) that any one example helps on average. In turn, this property permits us to reason directly about the CG explanation.

**Insights on Coherent Gradients (CG).** Based on our measurement of coherence in the setting of Zielinski et al. (2020), we can now pinpoint precisely what is missing in the original CG explanation for generalization: CG assumes that coherence exists, and from that, it shows how gradient descent can exploit it—this is the causal mechanism that the CG papers verify through intervention experiments.

**However, CG does not address where the coherence comes from and simply assumes that it is a property of the dataset. For example, CG assumes that real data has high coherence, and that data with random labels has low coherence (e.g., Section 2.2 in Chatterjee (2020)). Our measurements show that this assumption about coherence is not entirely true: For both real and random labels, at the start of training, coherence is low, and then increases.**

That said, consistent with the intuition from CG, we find that the peak coherence with real labels is much higher (each example helps $\approx 10^4$ other examples) than the peak with random labels (each example only helps $\approx 10^2$ other examples). Furthermore, the peak with real labels is reached very quickly (well within the first epoch) whereas with random labels it takes much longer (many epochs).

The failure of this assumption leads to a very interesting open question:

> **What causes the alignment of per-example gradients, i.e., coherence to increase in the course of training?**

Since the increase happens both in the case of real labels and in the case of random labels, we conjecture that the increase is an optimization phenomenon and not directly related to generalization. Going back to the analogy with random forests in the introduction, the creation of coherence is similar to the finding of commonality (possibly spurious) between examples during decision tree construction (c.f. discussion on generalization in deep learning in Section 4 of Chatterjee & Mishchenko (2020)).

In other words, while an explanation for why gradient descent generalizes is provided by the original CG hypothesis (if coherence exists, gradient descent exploits it by taking a larger step in the direction that improves many examples), we now need an explanation for the optimization phenomenon of why coherence increases in gradient descent (since it does not start off high even in the case of real data).

It is important to note that in contrast with the *increase* in coherence which is a mystery, the *decrease* in coherence as training progresses, and more examples get fit, is expected, as may be observed from Lemma 4 and the example on Coherence Reduction.

**Practical Applications.** Finally, it would be interesting to explore some practical applications of this metric. For example, we could consider using a metric derived from $m$-coherence to predict the generalization gap as a result of training. While a simple heuristic might be to use the area under the curve of $m$-coherence over time as an (inverse) measure of the generalization gap, a better heuristic would also incorporate for the distance travelled in parameter space since, as our experiments show, low coherence for relatively few steps (e.g., early in training with real labels) or when gradients are small (e.g., late in training) is not fatal for generalization.

Another possible use of $m$-coherence could be to freeze certain layers if they have low coherence to prevent overfitting in those layers or to improve the speed of training (e.g. if all lower layers are frozen). These practical applications would require even faster methods to estimate coherence. One way to do that might be to compute the coherence between mini-batches (with exponential moving averages) and then applying Corollary 3.1 in reverse.

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

# A    OMITTED PROOFS

We present the proofs omitted from Section 4. Recall from (4) that

$$\alpha(\mathcal{V}) = \frac{\underset{v\sim\mathcal{V},v'\sim\mathcal{V}}{\mathbb{E}}[v \cdot v']}{\underset{v\sim\mathcal{V}}{\mathbb{E}}[v \cdot v]}$$

**Theorem 1** (Boundedness). *We have $0 \leq \alpha(\mathcal{V}) \leq 1$. In particular, $\alpha(\mathcal{V}) = 0$ iff $\mathbb{E}_{v\sim\mathcal{V}}[v] = 0$ and $\alpha(\mathcal{V}) = 1$ iff all the vectors are equal.*

*Proof.* Since $v \cdot v \geq 0$ for any $v$, we have $\mathbb{E}_{v\sim\mathcal{V}}[v \cdot v] \geq 0$. Furthermore, it is easy to verify by expanding the expectations (in terms of the vectors and their corresponding probabilities) that

$$\underset{v\sim\mathcal{V},v'\sim\mathcal{V}}{\mathbb{E}}[v \cdot v'] = \underset{v\sim\mathcal{V}}{\mathbb{E}}[v] \cdot \underset{v\sim\mathcal{V}}{\mathbb{E}}[v] \geq 0. \tag{6}$$

Therefore, $\alpha(\mathcal{V}) \geq 0$. Likewise, another direct computation shows that

$$0 \leq \underset{v'\sim\mathcal{V}}{\mathbb{E}}\left[(\underset{v\sim\mathcal{V}}{\mathbb{E}}[v] - v') \cdot (\underset{v\sim\mathcal{V}}{\mathbb{E}}[v] - v')\right] = \underset{v\sim\mathcal{V}}{\mathbb{E}}[v \cdot v] - \underset{v\sim\mathcal{V}}{\mathbb{E}}[v] \cdot \underset{v\sim\mathcal{V}}{\mathbb{E}}[v] \tag{7}$$

Since from Equation 6 we have $\mathbb{E}[v] \cdot \mathbb{E}[v] = \mathbb{E}[v \cdot v']$, it follows that $\alpha(\mathcal{V}) \leq 1$. Furthermore, since each term of the expectation on the left is non-negative, equality is attained only when all the vectors are equal. □

**Lemma 2** (Scale Invariance). *For non-zero $k \in \mathbb{R}$, let $k\mathcal{V}$ denote the distribution of the random variable $kv$ where $v$ is drawn from $\mathcal{V}$. We have $\alpha(k\mathcal{V}) = \alpha(\mathcal{V})$.*

*Proof.*

$$\alpha(k\mathcal{V}) = \frac{\underset{v\sim k\mathcal{V},v'\sim k\mathcal{V}}{\mathbb{E}}[v \cdot v']}{\underset{v\sim k\mathcal{V}}{\mathbb{E}}[v \cdot v]} = \frac{\underset{v\sim\mathcal{V},v'\sim\mathcal{V}}{\mathbb{E}}[kv \cdot kv']}{\underset{v\sim\mathcal{V}}{\mathbb{E}}[kv \cdot kv]} = \frac{\underset{v\sim\mathcal{V},v'\sim\mathcal{V}}{\mathbb{E}}[v \cdot v']}{\underset{v\sim\mathcal{V}}{\mathbb{E}}[v \cdot v]} = \alpha(\mathcal{V}) \tag{8}$$

□

**Theorem 3** (Stylized mini-batching). *Let $v_1, v_2, .., v_k$ be $k$ i.i.d. variables drawn from $\mathcal{V}$. Let $\mathcal{W}$ denote the distribution of the random variable $w = \frac{1}{k}\sum_{i=1}^{k} v_i$. We have,*

$$\alpha(\mathcal{W}) = \alpha(k\mathcal{W}) = \frac{k \cdot \alpha(\mathcal{V})}{1 + (k-1) \cdot \alpha(\mathcal{V})} \tag{9}$$

*Proof.* The first equality follows from Lemma 2. For the second equality, we have,

$$\alpha(k\mathcal{W}) = \frac{\underset{\substack{w\sim k\mathcal{W},\\w'\sim k\mathcal{W}}}{\mathbb{E}}[w \cdot w']}{\underset{w\sim k\mathcal{W}}{\mathbb{E}}[w \cdot w]} = \frac{\underset{\substack{v_1,..,v_k,\\v'_1,..,v'_k}}{\mathbb{E}}[(\sum_i v_i) \cdot (\sum_i v'_i)]}{\underset{v_1,..,v_k}{\mathbb{E}}[(\sum_i v_i) \cdot (\sum_i v_i)]} = \frac{k^2 \underset{\substack{v\sim\mathcal{V},\\v'\sim\mathcal{V}}}{\mathbb{E}}[v \cdot v']}{k\underset{v\sim\mathcal{V}}{\mathbb{E}}[v \cdot v] + k \cdot (k-1)\underset{\substack{v\sim\mathcal{V},\\v'\sim\mathcal{V}}}{\mathbb{E}}[v \cdot v']}$$

By dividing the numerator and denominator of the last expression by $k \underset{v\sim\mathcal{V}}{\mathbb{E}}[v \cdot v]$ the required result follows. □

**Corollary 3.1** (Minibatch amplification). *$\alpha(\mathcal{W}) \geq \alpha(\mathcal{V})$ with equality iff $\alpha(\mathcal{V}) = 0$ or $\alpha(\mathcal{V}) = 1$.*

*Proof.* From the previous theorem, the transformation in coherence due to stylized mini-batching is given by the map $\alpha \mapsto \frac{k\cdot\alpha}{1+(k-1)\cdot\alpha}$. Now, since $\alpha \leq 1$, we have $k \geq 1 + (k-1) \cdot \alpha$, and since $\alpha \geq 0$, multiplying both sides by $\frac{\alpha}{1+(k-1)\cdot\alpha}$ we have $\frac{k\cdot\alpha}{1+(k-1)\cdot\alpha} \geq \alpha$. Finally, it is easy to check that the only two fixed points of the map are $\alpha = 0$ and $\alpha = 1$. □

**Lemma 4** (Effect of zero gradients). *If $\mathcal{W}$ denotes the distribution where with probability $p > 0$ we pick a vector from $\mathcal{V}$ and with probability $1 - p$ we pick the zero vector then $\alpha(\mathcal{W}) = p \cdot \alpha(\mathcal{V})$.*

*Proof.*

$$\alpha(\mathcal{W}) = \frac{\underset{w \sim \mathcal{W}, w' \sim \mathcal{W}}{\mathbb{E}} [w \cdot w']}{\underset{w \sim \mathcal{W}}{\mathbb{E}} [w \cdot w]} = \frac{p^2 \cdot \underset{v \sim \mathcal{V}, v' \sim \mathcal{V}}{\mathbb{E}} [v \cdot v']}{p \cdot \underset{v \sim \mathcal{V}}{\mathbb{E}} [v \cdot v]} = p \cdot \alpha(\mathcal{V}) \qquad (10)$$

$\square$

## B  ADDITIONAL EXPERIMENTAL RESULTS AND DISCUSSION

**Experimental Setup.** Our code for running experiments was heavily based on an open source Tensorflow example,[5] with modifications to allow label randomization and coherence metric logging. We used SGD with momentum (0.9), a batch size of 4096, and the learning rate schedule proposed in Goyal et al. (2017). We did not use weight decay or random augmentation of the input.

When measuring per-example gradients to compute $m$-coherence, for normalizing activations in a batch norm (BN) layer, we use the moving averages collected by the layer, i.e., we measure per-example gradients in eval mode. However, in the first few steps of training, since the moving averages are initialized at 0, the per-example gradients measured in this manner may not be accurate. We address this by priming the moving averages in the BN layers by running 40 steps without updating any trainable parameters (or momentum values) with a *no-op optimizer* at the start of our experiments.

The importance of this correction depends upon the architecture. For example, for ResNet-18 on ImageNet, without this correction, we see an artificially high loss (16 instead of the expected $\ln(1000) \approx 6.9$) in the first few steps of training due to very large activations feeding the final softmax layer. This leads to artificially high gradient measurements and $m$-coherence. However, these measurement artifacts disappear in about 25 steps as the BN averages get primed.

**Variation across Layers.** One advantage of $m$-coherence is that it is natural to use it to compare different projections of the per-example gradients and as such can be used to directly compare different layers with each other. Rows 4, 5 and 6 of Figure 1 shows the $m$-coherence of the classification layer, a convolution layer in the middle and the first convolution layer respectively. We only show the $m$-coherence for the weights (since in a spot check the $m$-coherence for weights and biases for a given layer looked very similar). We make a few observations.

First, convolutional layers have higher coherence than the fully connected layer. Although this could be a function of depth, we note that this may also be expected for a different reason. Since convolutional layers have filters that are instantiated at multiple sites, and the gradients from those sites for a single example add up in the overall gradient for a single gradient. Therefore, by reasoning similar to that of Lemma 3.1, we expect the coherence of the gradient across sites to be greater than those for the sites individually. And with more sites, we expect greater coherence. This is another way to see that weight sharing prevents overfitting.

Second, the convolutional layers, particularly the first one shows high coherence for random labels (though still generally lower than those for real labels). However, in the fully connected layer, there is a much greater difference between real and random labels. For random, it is generally quite low reaching 10 at the peak, whereas for real, while the examples are being fit, it is usually above 10 and in early epochs, before overfitting sets in, it even exceeds 100.

Third, the only place where $m$-coherence sometimes falls below 1 (the orthogonal limit) is in the first convolutional layer for real data *after* training accuracy has reached 100%. At that point, the layer may be over-constrained, i.e., improving the loss on one example may degrade it on another (though see the discussion below on test set coherence). Everywhere else, $m$-coherence tends to be at or above the orthogonal limit in line with our expectation that this learning problem is over-parameterized.

Finally, we note that the different layers for EfficientNet in Figure 3 show similar characteristics as those for ResNet.

---

[5]https://github.com/tensorflow/tpu/tree/master/models/experimental/resnet50_keras

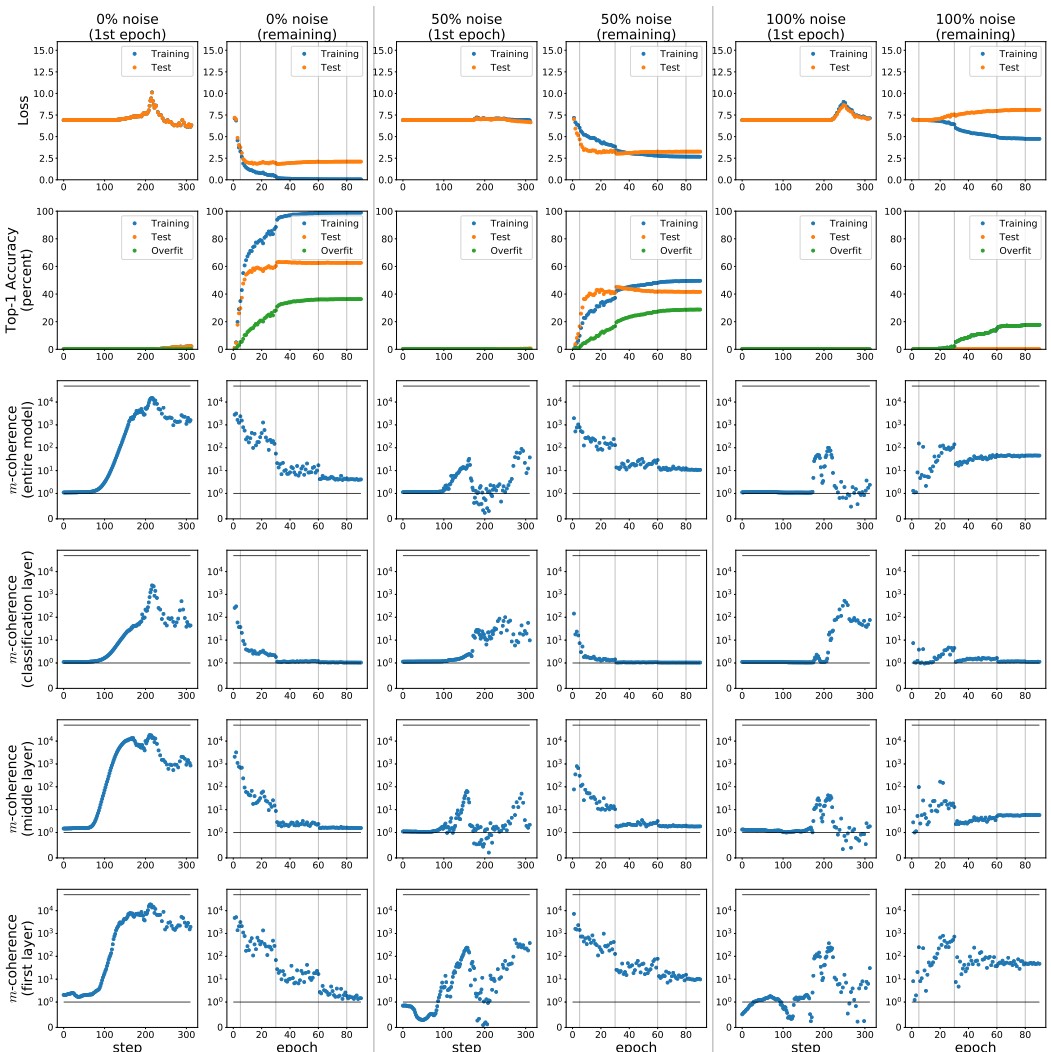

Figure 3: The evolution of alignment of per-example gradients of a EfficientNet-B0 network during training as measured with $m$-coherence on samples of size $m = 50{,}000$ on 3 variants of ImageNet with different amounts of label noise. We note that the results are qualitatively in agreement with what we see on ResNet, i.e., in both real and random cases we see coherence increase. A high peak is reached rapidly with real labels (in the first 150 steps) and a much lower peak is reached slowly with random labels (over many epochs). Horizontal lines for $m$-coherence are shown at 1 (the orthogonal limit) and at $m$. Vertical lines indicate sharp reductions in learning rate.

**Coherence on Test Set.** For completeness, we also measured the coherence on $m$ examples not used for training (drawn from the ImageNet validation set). They are shown in columns 1 and 2 of Figure 4 as "test." In the first epoch, we find that test and training coherence are roughly similar. However, when we look at the rest of training, we find that in the early part of the rest, test coherence is below that of training coherence, but in the later part, the opposite is true. This may be further evidence that coherence creation is a pure optimization phenomenon (as per the discussion in Section 6 of the main paper), i.e., the coherence creation (and subsequent consumption) is specific to the training examples.

It is interesting to observe that particularly for the convolutional layers, at the end of training, the test $m$-coherence is at 10 whereas training $m$-coherence is at 1 or even lower. This suggests that those layers are adapted to the training examples beyond what is likely to generalize.

**Reconciliation with Other Studies.** Fort et al. (2020) use the cosine and sign stiffness measures to study how gradient alignment depends on class membership, distance in input space between data

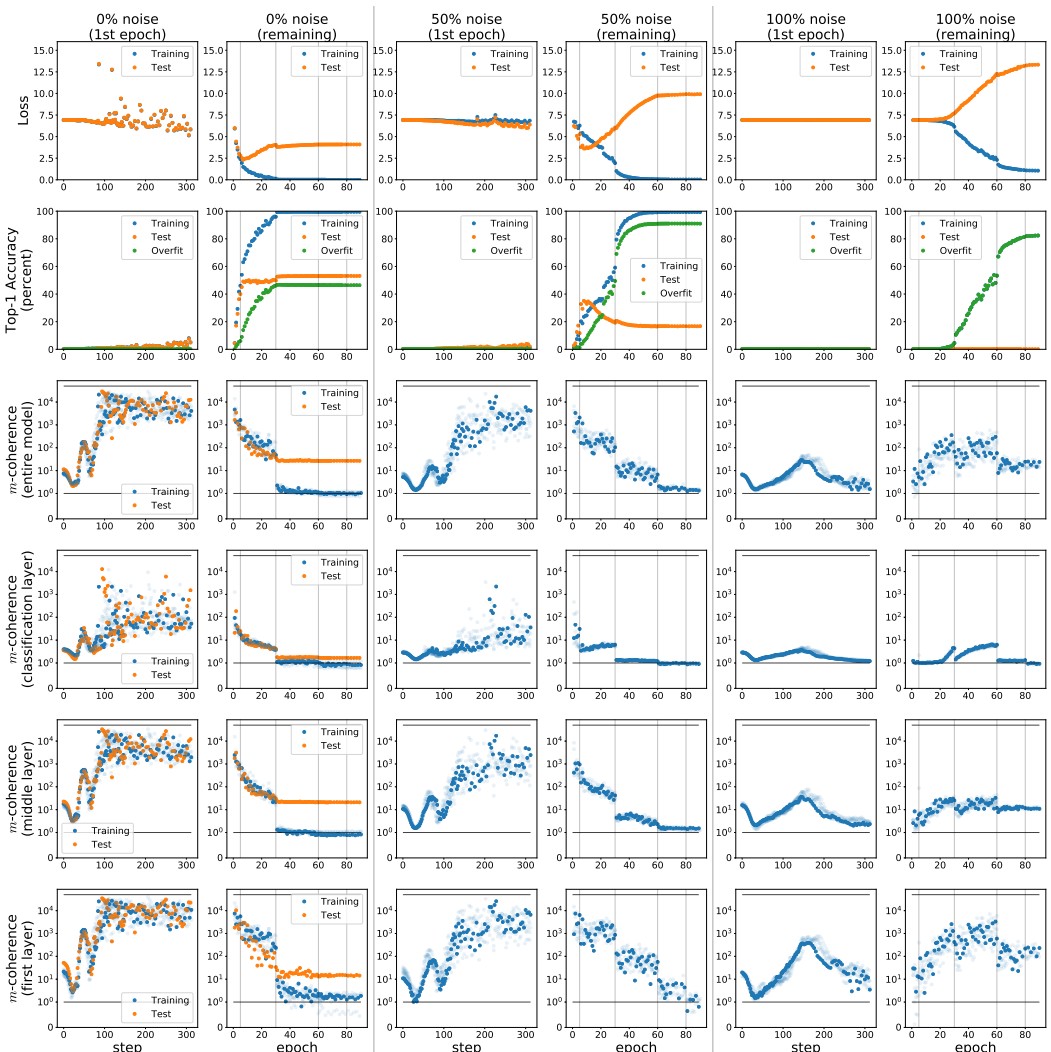

Figure 4: The evolution of alignment of per-example gradients of a ResNet-18 during training as measured with $m$-coherence on samples of size $m = 50{,}000$ on 3 variants of ImageNet with different amounts of label noise. This figure is similar to Figure-1, but also includes $m$-coherence plots for ImageNet test dataset (in column 1 and 2).

points, training iteration and learning rate. They use MNIST, Fashion MNIST, CIFAR-10/100 and MNLI datasets. Typical sample sizes are around 500 (for the 10 label datasets) and 3000 (for the 100 label datasets). They do not study label noise or memorization explicitly. In their class-based analysis, they find that initially, an example of a class only helps other examples in its class and adversely impacts examples of other classes. However, in the course of training, this effect goes down, and stiffness between classes goes up (though only to end up at 0).

We do not explicitly perform a class-based analysis, since with 1000 classes and about 1.2M training examples in ImageNet, we expect on average only 2 to 3 examples in each class pair. However, implicitly, our study is an inter-class analysis (though *not* a class-pair analysis) since in our sample, each example is expected to see roughly 1000 times as many examples of other classes as it does its own class. Our results indicate that examples in one class do help examples in other classes at different points in training since $m$-coherence is often in 1000s, and in a sample of approximately $50k$, we expect only about 50 examples per class.

However, since our metric is very different (as discussed in detail in Sections 2 and 3 of the paper), and the error bars in their study are large (as indicated in their Figure 5), we do not directly see any contradictions in the experimental data between their study and ours. Finally, we do not study

coherence as a function of input distance between examples or of learning rate changes, though we are interested in investigating the latter in future work.

Sankararaman et al. (2019) show theoretically that high gradient confusion impedes the convergence of SGD, and also analyze how factors such as network depth and width and initialization impact gradient confusion. They validate their theoretical results with experiments on MNIST, CIFAR-10, and CIFAR-100 (real labels only, since they do not study memorization) where they measure the minimum cosine similarity (MCS) between different training examples.[6] They mainly focus on the MCS value at the end of training as various architectural parameters are varied, but in Figures 7(c) and 8(c) in the appendix, they show the trajectory during training. There, we find that MCS starts low, increases to a peak and then comes back down again, in qualitative agreement with our findings.

Finally, in Figure 5, we show the individual terms of $\alpha$ (the numerator $\mathbb{E}[g_z \cdot g_{z'}]$ and denominator $\mathbb{E}[g_z \cdot g_z]$). Although the numerator can be estimated from the slope of the loss curve (as per equation (1) in the main paper), without the denominator to give it scale, it is hard to understand what variations are meaningful. As an extreme example, we see that, as expected from the loss curve (and this may be seen in the loss plots from other studies such as Zhang et al. (2017); Zielinski et al. (2020)), in the 100% random case, for the first 20 epochs or so, the norm of the expected gradient is close to zero (Figure 5, last column, row 1). However, there is significant activity in the denominator (row 2; and this is not typically recorded in experiments). By considering the quotient, and furthermore, *by putting it into context with the orthogonal limit* (as we do with $m$-coherence where that limit sets the scale), we can see that there is a definite build up in coherence in that period (row 3).

**Effect of sample size $m$.** Figure 6 shows the effect on $m$-coherence and $\alpha$ of doubling $m$ from our baseline value of 50,000 to 100,000. We see that the numerical values are generally the same, showing a slightly upward bias in $m$-coherence for the larger value of $m$ as might be expected.

**25% and 75% label noise.** The data for ResNet-18 training on 25% and 75% label noise is shown in Figure 7. This confirms the pattern noted in the main paper (Section 5) that with increasing noise, there is a decrease in the rate at which coherence increases.

---

[6]Since it is computed from mini-batch gradients rather than individual examples, it may over-estimate the gradient alignment (with the estimate being more inaccurate when alignment is low). Please see discussion in Section 3 of the main paper and Corollary 3.1.

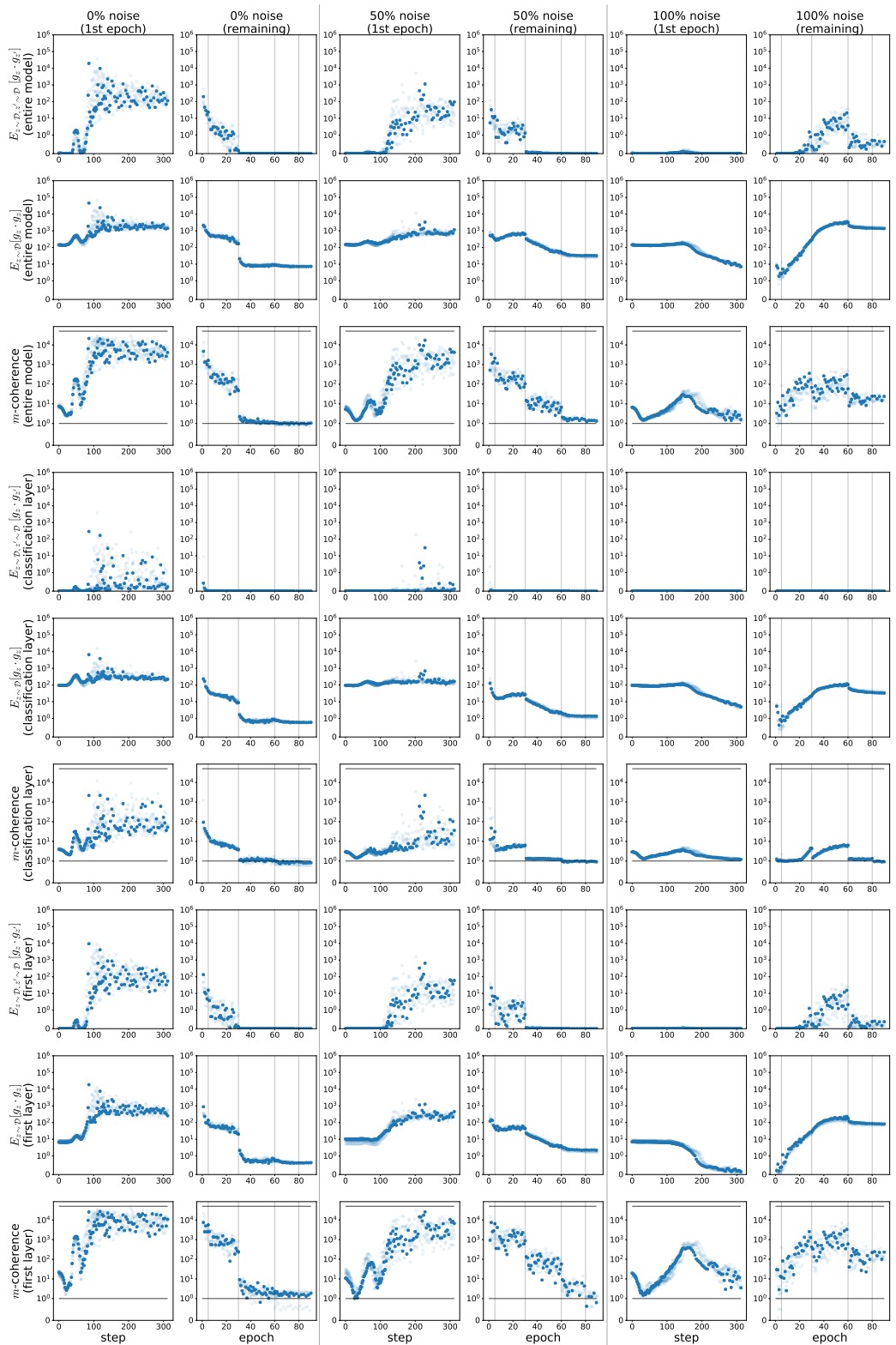

Figure 5: The expected gradients in the numerator and denominator for $\alpha$ (*not* $m$-coherence) corresponding to Figure 1. Note that even when the expected gradient is flat (as may be inferred even from the slope of the loss function), there is activity in the denominator which gets picked up with $\alpha$ or $m$-coherence particularly, if the scale is set appropriately w.r.t. to the orthogonal limit.

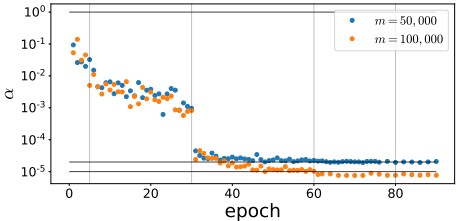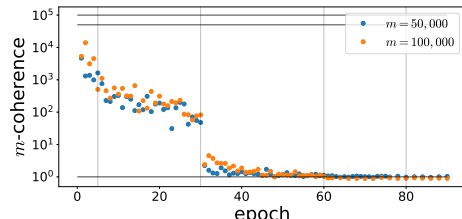

Figure 6: To understand the effect of $m$ for the values of $\alpha$ and $m$-coherence, we plot these values for $m = 50{,}000$ and $m = 100{,}000$ for the ResNet-18 training on 0% noise. In both plots we show horizontal lines for the orthogonal limit (which is different for the two samples in the $\alpha$ plot since it is $1/m$, but the same in the $m$-coherence plot since it is 1 in both cases) and the perfect alignment case (which is the same in the $\alpha$ plot since it is 1, and is different in the $m$-coherence plot since it is $m$.)

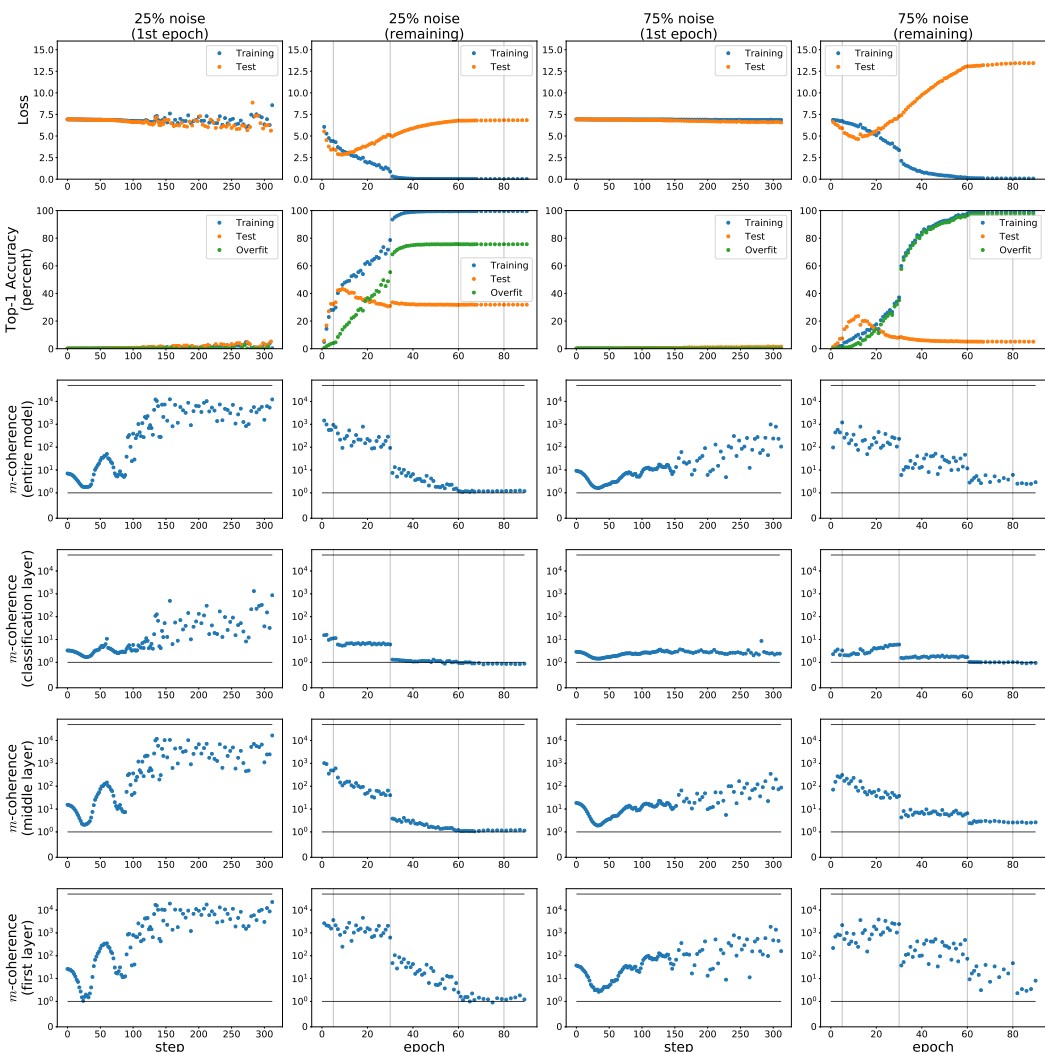

Figure 7: Evolution of $m$-coherence for 25% and 75% label noise (under the same settings as Figure 1). This confirms the pattern discussed in the main text that with increasing noise, the rate at which coherence is created in early training slows down.