# OpenReview forum: "Making Coherence Out of Nothing At All: Measuring Evolution of Gradient Alignment"
_ICLR.cc/2021/Conference — Reject_

### Official Review · AnonReviewer4 · 2020-10-22
**Official Blind Review #4**

**Rating:** 5
**Confidence:** 3

**Review:**

In this paper, the authors provide a new metric named m-coherence to measure the gradient coherences, i.e., the alignment of per-sample gradients. The proposed metric is to normalize the expected pairwise dot  product by an expected gradient norm. The authors show that the proposed metric is computationally efficient, mathematically simple, and may help researchers to investigate the gradient coherence behaviors during the entire training process. The authors provide some simple mathematical explanations and characterizations of the properties of m-coherence, e.g., boundedness, relativeness to sample minibatch, the impacts of zero gradients. The empirical results based on m-coherence are mostly intuitive. For example, as more training samples are fitted, the gradient coherence will decrease because more per-sample gradients are close to zero. There is also some surprising results. For example, the experiments on random labels show that the gradient coherence does not go to the expected 1.

Strength:

(1) The proposed metric is computationally efficient, and can be of interest to researchers who aim to explore the generalization behaviors of SGD or other algorithms in training neural networks.

(2) The paper is well written and easy to follow. The empirical results seem reasonable.

Weakness:

(1) Some of these empirical results are not surprising (and such results can be also easily made based on some other metrics, of course with additional cost). For me, it is more interesting to provide a completely theoretical demonstration via recent advances on theory of over-parameterized neural network rather than explain these results via merely looking at an empirical metric.

(2) Although the paper provides some surprising findings, e.g., experiments on random labels, this paper fails to provide an explanation on such findings, which, in my opinion, is more interesting. However, it seems to me that the current metric or tool cannot achieve this goal.

(3) One motivation of the new metric is the computational cost. Therefore, it would be better to demonstrate such an advantage over other metrices in the experiments. In addition, I am wondering whether the proposed metric is looser than other metrics (of course with higher computational cost), e.g., those in Fort et al., 2020 and Sankararamen et al., 2019. If so, it would be better to check whether such tightness will affect the empirical observations?

Overall, I am not excited about this paper, especially about its empirical results. For such reasons, I tend to weakly reject this paper. Since I am not an expert in this area, I am open to change my mind after reading other reviewers’ comments.

---

> ### Author Response · Authors · 2020-11-11
> **Why we are excited about the experimental results**
>
> Thank you for your review. As you and the other reviewers noted, this paper presents a new metric for gradient alignment that is mathematically cleaner, much more interpretable and easier to compute.
>
> From our experiments using this metric, we believe that the biggest takeaway is that we can pinpoint quite precisely what is missing in the original Coherent Gradients (CG) explanation for generalization ([ICLR 20](https://arxiv.org/abs/2002.10657) and [arXiv:2003.07422](https://arxiv.org/abs/2003.07422)). It is the following:
>
> > CG assumes that coherence exists, and from that it shows how Gradient Descent can exploit it (this is the causal mechanism that the CG papers verify). However, CG does not address where coherence comes from. It simply assumes that real data has high coherence, and that data with random labels have low coherence. Our paper shows that this is not entirely true. In both cases, at the start of training, coherence is low, and then increases.
>
> Consistent with the CG hypothesis, we do find that the peak coherence with real labels is much higher than in the case of random labels and this peak is reached very quickly in training. And that we see the same qualitative behavior in real labels and random labels is also an interesting observation since that means that the same mechanism can explain the range of behavior from good generalization to pure memorization.
>
> This leads to a very interesting open question for the field in our opinion:
>
> > What causes coherence to increase?
>
> Furthermore, since this increase happens both in the case of real data and in the case of random data (where there is nothing to learn), we believe this is an optimization phenomenon, i.e., it is not directly related to generalization.
>
> In other words, an explanation for why gradient descent generalizes is provided by the original CG hypothesis (if coherence exists, gradient descent exploits it by taking a larger step in the direction that improves many examples); but we as a community now need to explain why coherence increases in gradient descent (since it does not start off high even with real data).
>
> Does this address your concern (and if so would you recommend we pull this into a separate Conclusions sections to separate from the commentary in Section 6)? Looking forward to hearing from you.

---

> > ### Comment · AnonReviewer4 · 2020-11-22
> > **My concerns are not resolved**
> >
> > First thank the authors for their clarifications. However, the response seems to mainly claim that the metric is useful and the findings are interesting. I agree with this surely, but I think a more interesting thing is to explain these findings and provide some theoretical justifications. However, I do not get such points from their papers and response.
> >
> > In addition, the authors have not replied to my concern in (3) (i.e., about the tightness comparison among all metrics).
> >
> > For these reasons, I keep my score as 5.

---

> > > ### Author Response · Authors · 2020-11-22
> > > **We are not sure what you mean by "tightness comparison among all metrics"**
> > >
> > > Thank you for your response.
> > >
> > > ---
> > >
> > > This is mostly an empirical paper though we do believe we have theory around the metric that allows for more principled reasoning about what the measured values mean -- please see Sections 2 and 3.
> > >
> > > We cannot theoretically explain why coherence increases in early training at this point, but we do not think it is fair to hold that against the paper. It appears to be a very difficult question to answer on theoretical grounds, but one that is critical to understanding generalization.
> > >
> > > However, our theory can help answer why coherence must come down as examples are fitted (please see Lemma 4), and the theory of Coherent Gradients (that has been validated through other means) also provides some theoretical grounding to believe why the peak values are what they are in the different cases.
> > >
> > > ---
> > >
> > > Regarding your comment on "tightness comparison among all metrics". We do not understand what you mean by this.
> > >
> > > Gradient alignment is an intuitive notion (although one that is critical to understanding generalization as seen in Fort et al., Chatterjee, Zielinski et al., etc.) and there are several quantities one can measure to quantify this intuitive notion. For example, one could measure expected pairwise dot product of gradients, stiffness, gradient confusion, or the metric we propose $m$-coherence.
> > >
> > > Since there is no golden quantity for gradient alignment w.r.t. which we can quantify tightness, there is  there is no such tightness comparison in the paper.
> > >
> > > Finally, as discussed in Sections 2 and 3, expected pairwise dot products, stiffness and gradient confusion suffer from several defects (either not normalized, or ad hoc normalized; hard to reason about; difficult to compute; difficult to interpret) and thus we claim that $m$-coherence is the superior metric to quantify the intuitive notion of gradient alignment, and we hope that is the one that will be used going forward.
> > >
> > > We hope this addresses your concern.

---

### Official Review · AnonReviewer2 · 2020-10-26
**Interesting Empirical Results, Incremental and Misleading Explanations**

**Rating:** 5
**Confidence:** 3

**Review:**

In this work, the authors propose m-coherence as an replacement of sign stiffness / cosine stiffness / gradient confusion to measure the coherence of gradients in SGD training. Compared to existing measurements, m-coherence is more scalable and well-normalized. Using the m-coherence, the authors perform empirical studies on SGD optimizing a ResNet on ImageNet with natural or random label. The experiments show that for both real label and random label datasets,  the initial coherence is low, then increases as the training goes on, finally decreases as more and more data gets fitted.

# Interesting empirical results.
- I think the empirically similarity of gradient coherence for SGD learning real label and random label datasets is interesting. It suggests (1) the optimization of real label dataset and random label dataset is not totally different ---- they only differ in terms of fitting time; (2) we should look closer at the initial optimization of real label dataset though, since the "coherence increasing" phase in this case is super fast.

- On the other hand, the observation is not too surprising since even real label dataset contains noise, and random label dataset is only an extreme case where the noise in the dataset is super large.

- In sum, I personally do not think this pure finding, without an insightful analysis, is significant enough for an acceptance.

# Misleading interpretation.

- I think the writing of this paper is quite misleading. For example, in page 6 Real Labels paragraph, it claims for real labels, the initial coherence is very high? But then in page 7 Early Training paragraph, it instead claims the initial coherence is not very high if you look closer. Personally I find it misleading to make a claim first but then correct the claim with another one which is totally contradictory to the formal. I think it is better to explicitly point out the claims are for different scenarios, i.e., in large timescale/small timescale.

- Moreover, in page 8, the last part, Separation of Generalization and Optimization. The statements are super vague and not supported by experiments/theory. I do not understand why these arguments can make sense. I strongly encourage the authors to backup their statements with evidences.


# Experiments
- The authors claim several benefits of the proposed m-coherence compared with existing measurements like sign stiffness / cosine stiffness / gradient confusion. Indeed in terms of computation, m-coherence can be cheaper. But it is not clear whether or not sign stiffness / cosine stiffness / gradient confusion agrees with m-coherence empirically. In my opinion there should be some ablation studies to check the trend of gradient coherence with other measurements.

- Looking at Fig. 1, I find the plots are not consistent in format. For example, in the third row, the first two plots have test m-coherence, while the others do not have this; in the second row, the last plots misses a training accuracy curve; and so on. Please explain why.

# Minors

- Page 3, below Eq. (2). I guess it should be a "lower bound" instead of "upper bound"? Since in Eq. (2) there is a negative sign.

- Page 2, second paragraph last sentence. I am not sure about this claim. If the loss functions are not linear and gradient directions change every iterates (even for the same data point), the claim can be wrong. Because even at every iteration the gradients are orthogonal, abandoning one data point does not imply GD/SGD cannot optimize the direction along the gradient on this data, since other data points can contribute this direction afterwords.

---

> ### Author Response · Authors · 2020-11-16
> **Clarification on main experimental take-away and responses to detailed points**
>
> Thank you for your review.
>
> We would like to emphasize that the key contribution of this paper is a new metric $m$-coherence that is mathematically cleaner and much more interpretable than previous proposals (stiffness, confusion, etc.). This cleanliness allows us to reason theoretically about this metric and interpret the observations directly.
>
> On the empirical side, we believe that the biggest takeaway from our paper is that we can pinpoint quite precisely what is missing in the original Coherent Gradients (CG) explanation for generalization ([ICLR 20](https://arxiv.org/abs/2002.10657) and [arXiv:2003.07422](https://arxiv.org/abs/2003.07422)). It is the following:
>
> > CG assumes that coherence exists, and from that it shows how Gradient Descent can exploit it (this is the causal mechanism that the CG papers verify). However, CG does not address where coherence comes from. It simply assumes that real data has high coherence, and that data with random labels have low coherence. Our paper shows that this is not entirely true. In both cases, at the start of training, coherence is low, and then increases.
>
> Consistent with the CG hypothesis, we do find that the peak coherence with real labels is much higher than in the case of random labels and this peak is reached very quickly in training. And that we see the same qualitative behavior in real labels and random labels is also an interesting observation since that means that the same mechanism can explain the range of behavior from good generalization to pure memorization.
>
> This leads to a very interesting open question for the field in our opinion:
>
> > What causes coherence to increase?
>
> Furthermore, since this increase happens both in the case of real data and in the case of random data (where there is nothing to learn), we believe this is an optimization phenomenon, i.e., it is not directly related to generalization.
>
> In other words, an explanation for why gradient descent generalizes is provided by the original CG hypothesis (if coherence exists, gradient descent exploits it by taking a larger step in the direction that improves many examples); but we as a community now need to explain why coherence increases in gradient descent (since it does not start off high even with real data).
>
>
> ## Interpretation ##
>
> We rewrote the discussion section to capture the above discussion.
>
> Timescales. The suggestion of explicitly referring to the timescales is a good one. We edited the section to remove “initial coherence” which is confusing and was overloaded and replaced it with explicit reference to when the measurement was taken. Please take a look and let us know if it is still confusing.
>
> Generalization and Optimization. Deleted and replaced with a new Discussion section that captures clearly what the main takeaway from the paper is (along the lines above).
>
> ## Experiments ##
>
> Please see the comparisons in the Appendix under “Reconciliation with Other Studies” and Figure 4 (submitted version) or Figure 5 (updated version) for numerator/denominator ablations. Also, in addition to the computational benefit as you point out, we believe a very strong reason to adopt $m$-coherence is for its interpretability and mathematical cleanliness (see discussion in Section 2 and 3). Given the lack of interpretability of stiffness, confusion, etc. it is hard to make sense of the observed values.
>
> Figure 1, second row last plot: The training accuracy curve is there, but is hidden by the overfit curve that is coincident on it (as expected since it is 100% label noise). Based on suggestion by reviewer #2, we removed the overfit line to de-clutter the graph even more.
>
> Figure 1, test curve for 0% noise: Since the test $m$-coherence is not directly relevant, and clutters the plot, we only computed and showed it for 0% noise (just in case the reader was curious). However, since it is not directly relevant, and only discussed in the appendix, we moved it to the appendix (as Figure 4).
>
> ## Minor ##
>
> Page 3, below Eq. (2) is indeed “lower bound”. Fixed.
>
> Page 2, second paragraph last sentence. As we note, this is only for intuition. We added an example to illustrate concretely what we mean.

---

> > ### Comment · AnonReviewer2 · 2020-11-21
> > **Thank you for the clarifications**
> >
> > Thank you for the clarifications. I agree with most of them.
> >
> > In my current evaluation, this paper is marginal between acceptance or rejection. I tend to reject a pure empirical paper, but this is mostly based on personal preference --- most of the solid concerns have been addressed.
> >
> > I thus increase the score to 5, and meanwhile decrease my confidence to 3.

---

> > > ### Author Response · Authors · 2020-11-22
> > > **Thank you for revising the score (and some theoretical highlights)**
> > >
> > > Thank you for your response and for revising the score.
> > >
> > > We understand your preference for something more than a purely empirical paper, but we think this is useful empirical data for future research in generalization based on a metric that is inspired by theoretical considerations as described in Sections 2 and 3.
> > >
> > > Due to the resulting mathematical cleanliness, we can reason about it in a principled manner. This is not possible with the previous, more ad hoc quantities to measure gradient alignment. Please consider the following examples:
> > >
> > > First, we can reason about how measuring coherence at the batch level (i.e. coherence between batches) can overstate the measured coherence. This is not just a theoretical issue -- please see discussion on Page 15 of the appendix comparing to Sankararaman et al. (2019) (particularly footnote 6) -- where now we can see that their measurements may have overstated coherence.
> > >
> > > Second, as remarked in the paper (page 5) Theorem 3 provides a nice perspective on the type of results proved in Yin et al.
> > > (2018) and Jain et al. (2018). When $\alpha \ll1/k$ but non-zero,
> > > creating mini-batches of size $k$ increases coherence almost $k$ times. But, when $\alpha \approx 1$ (i.e., low
> > > diversity) there is not much point in creating mini-batches since there is little room for improvement. The specific statement of Theorem 3 provides a clean way to do this reasoning.
> > >
> > > Third, Lemma 4, although simple allows us to at least explain one aspect of the evolution of coherence -- that as examples get fitted, coherence must go down. Note that this is only possible because of the mathematical properties of the metric we propose, and one cannot do the equivalent reasoning for stiffness or confusion.
> > >
> > > Therefore, we believe that this paper makes a non-trivial theoretical contribution to the literature.

---

### Official Review · AnonReviewer1 · 2020-10-28
**A nice, simple method for understanding gradient coherence**

**Rating:** 8
**Confidence:** 4

**Review:**

######################################################################

1.  Paper Summary


This work introduces m-coherence, a new method for understanding gradient coherence when training deep neural networks.  The authors then present theoretical properties of m-coherence (importantly scale invariance) and demonstrate that m-coherence can be computed in a computationally efficient manner.  The authors then compute m-coherence across steps/epochs when training ResNet-18 and EfficientNet on ImageNet without label noise, with 50% label noise, and 100% label noise.  The experiments demonstrate that (1) m-coherence matches intuition after an initial phase  (i.e. m-coherence is large and decreases over time for real data) (2) m-coherences increases for all datasets to a peak in the first 100 epochs before decreasing.

######################################################################

2. Strengths

2.1. m-coherence is a very natural metric for understanding gradient coherence in that it is scale invariant, easy to compute, and is easy to analyze mathematically.

2.2. The description of m-coherence and its properties is presented well in comparison to recent work.   Additionally, the proofs follow almost immediately from the definitions and highlight how m-coherence is a simple, intuitive framework for understanding gradient coherence.

2.3. As m-coherence can be computed in a computationally efficient manner (both in algorithm complexity and memory usage), the authors are able to analyze gradient coherence on modern convolutional networks trained on 50,000 examples from ImageNet.  Importantly, the authors present both step-wise m-coherence and epoch-wise m-coherence, which highlights the phenomenon of m-coherences increasing for the first few steps of SGD.  This empirical finding could be an important stepping stone for theoreticians in understanding optimization and generalization for over-parameterized models.

2.4. The authors also highlight that the initial increase in m-coherence is robust across a variety of architectures, label noise settings, and initializations/minibatch constructions.  This finding is rather surprising in that it occurs even on datasets with 100% label noise.

######################################################################


3. Limitations/Questions

3.1. I would have liked to see an explicit comparison of m-coherence with the other recent methods on a subset of CIFAR10.  In particular, does a similar trend arise of gradient coherence first increasing before decreasing under these other methods?  I read through the discussion in the appendix about this result being consistent across these papers, but there appear to be caveats with what samples were considered in other works.  It would be interesting to compare gradient coherence under these methods using a fixed random subset of CIFAR10 across methods (please let me know if I missed anything though in case this is already done).

3.2. (Minor) Just for some clarity, it would be interesting to understand the makeup of the 50,000 samples considered in the experiment.  How far away is the sample set from having 50 examples per class?  ImageNet also has a large number of classes of similar objects (e.g. ~100+ dog classes).  Is there indeed a higher m-coherence for such subsets of ImageNet during training or does m-coherence actually increase on these subsets as well?


######################################################################

4. Score and Rationale

Overall, I vote for accepting this paper.  I find the contribution of m-coherence to be novel, practical step towards understanding gradient coherence for modern machine learning settings.  In particular, I feel that these findings will be of interest to theoreticians interested in the intersection of optimization and generalization.  My only criticism of the work would be that the authors should try to provide a more concrete comparison with other works on a random subset of CIFAR10 to determine whether gradient coherence phenomena are robust across all these methods.


######################################################################


5. Minor Comments

5.1. I think the "overfit" label in Figure 1 is a bit confusing in the second row of Figure 1- is this just the gap between and test error? If so, I feel that just re-labelling this as the "Gen. Gap" or even just removing it would be fine.

---

> ### Author Response · Authors · 2020-11-16
> **Thank you**
>
> Thank you for your review and for the encouragement.
>
> **3.1** Thank you for the experiment suggestions. We are out of office at the moment but can try to run them when back. Due to the ad hoc normalizations employed in stiffness and gradient confusion (they ignore gradient magnitudes which breaks the connection with the loss function), we thought a direct comparison would not be particularly informative. But please note that in the Appendix (Figure 5) we show comparisons w.r.t. the pairwise expected dot product and the expected per example gradient norms which due to their direct relationship with the loss function are a bit more interpretable (though still hard to compare across architectures or timesteps since they are not normalized).
>
> **3.2** In the ImageNet training set (ILSVRC2012), most of the 1000 classes have 1300 examples each, though a few classes have less (but even the smallest class has over 700 examples) [https://arxiv.org/pdf/1409.0575v3.pdf, Table 2]. In each experiment run, we chose a different set of 50,000 examples uniformly at random from this training set, so we should expect 50 examples give or take per class.
>
> It is an interesting idea to look at the coherence across semantically related classes in the ImageNet label hierarchy but we have not looked at that yet.
>
> **5.1** Yes, that’s a good idea. We will remove the overfit line (since it obscured the training curve in the 100% noise case and confused a reviewer).

---

### Official Review · AnonReviewer3 · 2020-10-29
**The paper studies a new notion of coherence to explain generalizability of neural networks. Coherence is defined as the degree to which the gradients w.r.t the data points are aligned with each other. The paper presents experiments to study the evolution of the proposed metric during training and testing of neural networks.**

**Rating:** 6
**Confidence:** 3

**Review:**

The paper studies a new notion of coherence to explain generalizability of neural networks. Coherence is defined as the degree to which the gradients w.r.t the data points are aligned with each other. The proposed metric to measure coherence improves upon existing definitions in the literature by ensuring a natural normalization such that the metric is always between 1 and the number of data points, thus having a direct interpretation of the number of data points whose gradients are aligned.

## Pros:
1. The experiments presented are pretty interesting and the paper does a good job of interpreting the results of the experiments. Studying the evolution of the coherence can be seen to be insightful in understanding generalizability.
2. The proposed definition of the metric enjoys a better computational complexity, making it possible to experiment on larger datasets.
3. The paper also provides some general theory on coherence w.r.t to general sets, which is appreciated and manages to provide a better intuition to the readers.

## Cons
1. The general discussion in section 6. seems hand-wavy. Examples: 1)  "One may imagine an uneasy equilibrium between these opposing tendencies leading to expansion
and contraction in coherence. As soon as significant coherence builds up, it leads to an increase in the
effective learning rate (higher relative gradient norm) leading to faster consumption" - It's hard to say which one is the cause and which is the effect.  2) "Since this creation happens even with random labels where there is nothing to learn (i.e.,
no generalization), there is reason to believe that this creation is purely an optimization phenomenon."

2. In general, although section 6 adds value to the paper, it does not seem to be the result of a principled analysis and seems more like a commentary on the plots.


##  Overall score:
I am giving this paper an overall score of 6. The paper is interesting and the ideas presented are sound. My reasons for not providing a stronger recommendation for acceptance are
1. I am not convinced that there is one particular message that the paper finally arrives at. There is a new metric defined, and some general theory provided w.r.t this metric and experiments that study the evolution of the proposed metric. It would have been good to have a discussion how this particular set of experiments can better inform practice or theory of deep learning.
2. The analysis is vague and hand-wavy.

Some potential directions for improvement:
1. Studying how the coherence changes w.r.t the complexity of the model. This can maybe performed using toy datasets and small architectures. What happens when layers are added, removed, made wider, and so on.
2. Studying the evolution of coherence across different cross-validation folds to see if coherence can be used a way to tune hyper-parameters.
3. Study of what happens when there are adverserial examples.

---

> ### Author Response · Authors · 2020-11-11
> **Thank you and clarification on the main experimental takeaway**
>
> Thank you for your review. Section 6 is indeed meant to be an informal discussion/commentary on the plots (perhaps we should clarify that at the outset), and we would like to firm it up and distill it to something clearer based on the discussion here.
>
> Your comment about having a clear takeaway from the experimental results in the paper resonated with us. We believe that the biggest takeaway from our paper is that we can pinpoint quite precisely what is missing in the original Coherent Gradients (CG) explanation for generalization ([ICLR 20](https://arxiv.org/abs/2002.10657) and [arXiv:2003.07422](https://arxiv.org/abs/2003.07422)). It is the following:
>
> > CG assumes that coherence exists, and from that it shows how Gradient Descent can exploit it (this is the causal mechanism that the CG papers verify). However, CG does not address where coherence comes from. It simply assumes that real data has high coherence, and that data with random labels have low coherence. Our paper shows that this is not entirely true. In both cases, at the start of training, coherence is low, and then increases.
>
> Consistent with the CG hypothesis, we do find that the peak coherence with real labels is much higher than in the case of random labels and this peak is reached very quickly in training. And that we see the same qualitative behavior in real labels and random labels is also an interesting observation since that means that the same mechanism can explain the range of behavior from good generalization to pure memorization.
>
> This leads to a very interesting open question for the field in our opinion:
>
> > What causes coherence to increase?
>
> Furthermore, since this increase happens both in the case of real data and in the case of random data (where there is nothing to learn), we believe this is an optimization phenomenon, i.e., it is not directly related to generalization.
>
> In other words, an explanation for why gradient descent generalizes is provided by the original CG hypothesis (if coherence exists, gradient descent exploits it by taking a larger step in the direction that improves many examples); but we as a community now need to explain why coherence increases in gradient descent (since it does not start off high even with real data).
>
> Does this address your concern (and if so would you recommend we pull this into a separate Conclusions sections to separate from the commentary in Section 6)? Looking forward to hearing from you.

---

### Author Response · Authors · 2020-11-16
**New version uploaded with rewritten discussion to capture key experimental takeaways**

Thank you everyone for the detailed reviews.

We have just uploaded a new version of the paper that has a completely rewritten Discussion section to crisply articulate what the main takeaways from the experimental part of the paper are and why they are important.  The new version also has fixes for some presentation issues pointed out by the reviewers.

We have also responded individually to the reviewers. Look forward to hearing from all of you.

---

### Decision · Program_Chairs · 2021-01-07
**Final Decision**

**Decision:**

Reject

**Comment:**

The average score of the reviewers is 6. There are various pros and cons pointed out by the reviewers. Unfortunately, the AC and SAC found that the merit could be outweighed by the limitations of the work, and would like to recommend rejection. For example, a central concern raised by the reviewers is the lack of theoretical justification---the measurement that the paper proposes does seem to show an interesting empirical phenomenon, but as a few reviewers pointed out, it's unclear how the interesting phenomenon directly links to the generalization mechanism, and it's unclear whether the new interesting phenomenon is caused by the change of measurement of the gradient coherence or it is something fundamental. The arguments related to these are found to be generally vague and hand-wavey by the reviewers.  The AC would like to encourage the authors to address the reviewers' concerns thoroughly, especially those regarding the difference and similarity with prior works, the interpretation, and limitations of the results, etc., and consider adding more rigorous analysis to justify the proposed measurement of gradient alignment.